

# A new X-ray transparent flow-through reaction cell for a µ-CT-based concomitant surveillance of the reaction progress of hydrothermal mineral-fluid interactions

Wolf-Achim Kahl[1], Christian Hansen[1,2], Wolfgang Bach[1]

[1]Department of Geosciences, University of Bremen, Klagenfurter Straße (GEO), D-28359 Bremen, Germany
[2]now: Research Group for Marine Geochemistry (ICBM-MPI Bridging Group), Carl von Ossietzky University of Oldenburg, Institute for Chemistry and Biology of the Marine Environment (ICBM), 26129 Oldenburg, Germany

*Correspondence to*: Wolf-Achim Kahl (wakahl@uni-bremen.de)

**Abstract.** A new flow-through reaction cell consisting of an X-ray transparent semicrystalline thermoplastic has been developed for percolation experiments. Core holder, tubing and all confining parts are designed of PEEK (polyetheretherketone) to allow concomitant surveillance of the reaction progress by X-ray microtomography (µ-CT). With this cell setup, corrosive or oversaturated fluids can be forced through rock cores (up to Ø 19 mm) or powders at pressures up to 100 bars and temperatures up to 200 °C. The reaction progress of the experiment can be monitored without dismantling

the sample from the core holder.

The combination of this flow-through reaction cell setup with a laboratory X-ray µ-CT system facilitates on-demand monitoring of the reaction progress of (long-term) hydrothermal experiments in the own laboratory, keeping interruption times as short as possible. To demonstrate both the suitability of the cell construction material for X-ray imaging purposes and the experimental performance of the flow-through system, we report the virtually non-existent bias of the PEEK cell

setup with distinctive X-ray observations (e. g., differing states of pore fillings: air vs. fluid; detection of delicate fabric elements: filigree zeolitic crystals overgrowing weathered muscovite), and the monitoring of the gypsum/anhydrite transition as a case study of a 4D fabric evolution.

## 1 Introduction

Fluid-rock interactions govern the critical mass transfers involved in the exchange between the oceanic crust and seawater and the formation of hydrothermal deposits in a range of settings. They also influence hydraulic and rock mechanical properties and are hence relevant to applied geosciences. Understanding these processes requires consideration of multiple factors, including changes in porosity distribution, permeability and their relationship with critical dissolution and precipitation rates. To date, the knowledge of these interdependencies is very limited, and we are hence often inable to make

reliable predictions of fluid-rock behaviour in natural systems.



In numerical simulations of hydrothermal systems, fluid flow, heat- and matter transport, and chemical reaction progress have to be considered simultaneously (e.g., Kühn, 2009). The first step in understanding macroscopic rock properties, such as permeability or capillary pressure, is the correlation of rock microstructure and physical properties of the fluid at the pore

scale. A crucial part in this is the transformation of real pore space geometries into appropriate pore space models. In this regard, true 3D spatially resolved spectroscopic data (e.g., nuclear magnetic resonance; X-ray or synchrotron computed microtomography: µ-CT) is superior to pure statistical considerations of porous media, e.g. by mercury porosimetry (Øren and Bakke 2003). Recent studies demonstrated that structural heterogeneities on the pore-scale in rocks govern the hydrodynamic flow zones within porous media. In turn, these hydrodynamic flow zones control transport and reaction

processes and local and effective reaction rates (e. g., Andreani et al., 2009). Suitably, X-ray microtomography facilitates the non-invasive assessment of microfabrics in the µm-range. The non-destructive nature of µ-CT investigations is of paramount importance for monitoring reaction progress in long-term fluid-rock reaction experiments. Several studies have successfully employed µ-CT-derived pore space geometries and experimental data from percolation experiments with rock micro-cores to parameterize coupled transport-reaction models (e.g., Flukiger and Bernhard, 2009; Noiriel et al., 2009).

In a number of studies, reaction progress in hydrothermal fluid-rock interaction experiments was investigated by means of lab-based or synchrotron µ-CT: pore space evolution, changes in rock fabric and mineral surface area, as well as hydraulic scale displacement effects in percolation experiments were quantified by comparison of the initial and the post-reaction states. This procedure has been applied to a variety of geological materials and model rock analogues of various dimensions, e.g. micro-cores of sandstone (Sell et al., 2012: core Ø 47.6 mm, length 46.0 mm), limestone (Gouze & Luquot, 2011: core

Ø 9 mm, 18 mm length; Smith et al., 2013a: core Ø 15 mm, 30 mm length; Vialle et al., 2014: Ø 100 mm, length 350 mm), anhydrite (Smith et al., 2013b: core length 30 mm), limestone analogue (Noiriel et al., 2012: mixed calcite grains and glass beads, Ø 6.5 mm, length 12 mm).

All former studies have in common, however, that the samples were only scanned twice, prior to confinement within the reaction cell and at the end of the experiments after a complete dismantling. Reaction concomitant scans were not possible,

either because of X-ray intransparent reactor materials were used or because of impracticabilities in introducing the pressure housing into the beam chamber. In case of 4D µ-CT studies that rely on more than the initial and the final time step, removing the sample from the experiment, and its reassembly after scan, has to be mastered with minimal perturbation of the core. Burlion et al. (2006) performed leaching experiments on concrete samples in a batch reactor setup, removing the sample five times from the experiment for synchrotron X-raying. Noiriel et al. (2009) scanned a limestone sample before,

after 9h, and after 15h (end of experiment); for the scans the sample was removed, but remained water-saturated. Noiriel et al. (2005) surveyed limestone dissolution by comparing three scans, taken after subsequent stages of reactive percolation, with the unreacted sample (cores Ø 9 mm, length 21 mm; samples were removed from cell for the scans).



In each of these studies, the core was removed from its pressure housing prior to scanning so that disturbance of the core during dismantling / remounting for repeated scanning was an issue. This problem can be avoided, if the sample stays mounted for the scan. Likewise, image registration of subsequent scans can be simplified that way. Currently, cell materials with a low X-ray attenuation coefficient that allow the scan of a mounted sample, are deployed to image important

hydrodynamic  phenomena on the pore-scale (e.g., two-phase flow in porous media). Since high-speed tomography (even sub-second) has become possible at synchrotron facilities, in situ drainage / imbibition or two-phase flow in sample cores have been imaged in real-time (e. g., Myers et al., 2011: air-drainage of water-saturated Bentheimer sandstone, core Ø 5 mm, 12 mm length; Berg et al., 2013: imaging of pore-scale displacement events in Berea sandstone in real-time, core Ø 4 mm, 10 mm length). Most recently, Bultheys et al. (2015) were able to perform in situ, time-resolved imaging of dynamic water-

rock interaction processes using a laboratory-based X-ray micro-computed tomography scanner and a PMMA flow cell (drainage of water-saturated Bentheim sandstone by an oil phase: core Ø 6 mm, 17 mm length) to visualize the time-dependant development of preferential flow channels in limestone.

Here, we present a new flow-through reaction cell consisting of an X-ray transparent semicrystalline thermoplastic that has been developed for (long-term) percolation experiments to allow concomitant surveillance of the reaction progress by X-ray

microtomography without dismantling the sample. The unique material properties of the weakly attenuating, but mechanically strong, cell material allow the experimental examinaton of rock-fluid interactions under low to moderate temperatures (up to 200 °C) and fluid pressures up to ca. 100 bar (10 MPa).

## 2 Materials and methods

### 2.1 PEEK Percolation Cell


The semicrystalline thermoplastic material (polyetheretherketone: PEEK) is commercially available (KTK Kunststofftechnik Vertriebs GmbH, Germering, Germany). Since PEEK is machinable, withstands pressure and has a high mechanical strength and hardness over a broad temperature range, an external metal pressure vessel housing is not necessary in the experimental setup. Instead, 20 mm thick PEEK walls are sufficient to safely maintain pressures up to 100 bars (10 MPa) and temperatures

up to 200 °C. We use two PEEK modifcations for two different temperature ranges: PEEK extruded natural up to 120 °C, and PEEK mod extruded up to 200 °C. The percolation cell (Fig. 1) consists of a dumbbell-shaped central part (housing the core holder) and two end caps, which are assembled by screws connecting a set of metallic rings and half-rings. The assembled cell measures 210 mm in height and 100 mm in width. The central part of the dumbbell contains the core holder and is radiographed during the scan (Fig. 1D). It is composed of a hollow cylinder of 22 mm width with 20 mm thick walls.

Both in- and outlet plugs of the core holder are 19 mm in diameter and 30 mm long; they are connected to the PEEK capillary via PEEK 10-32 fingertight fittings.



The reaction cell can accomodate rock cylinders of 19 mm diameter and up to 50 mm length that are mantled with FEP (fluoroethylenepropylene; Adtech Polymer Engineering ltd., Stroud, UK) heat-shrink sleeves. PEEK capillary tubing (1/16" O.D. x 0.020" I.D.; 1.60 mm O.D. x 0.5 mm I.D.) provides fluid circulation through the percolation unit. Spiderweb-type groove patterns on the end-faces of the core holder's inlet and outlet plugs ensure an evenly distribution of the fluids over the entire face of the core before water enters or leaves the plug. An optional external application of additonal mantle pressure (overburden) on the heat-shrink sleeve ensures a well confined maximum fluid flow through the rock core, even at elevated experimental pressure levels.

The flow-through system (Fig. 1C; schematic flow diagram in Fig. 2) uses two injection pumps (Teledyne Isco D-Series syringe pumps), which can be run either independently of each other or in a synchronised mode. All pumps are of a pulsation-free, positive-displacement pump type and each offers an injection capacity of 500 cm3 in one injection cycle. Fluids discharged from cell are passed through a back-pressure regulator that maintains a constant pressure level inside the flow-through apparatus. All parts that are actually in contact with reaction fluids are either made of PEEK (core holder end plugs, capillary), FEP (shrink-sleeve) or titanium (tip of needle valves). During the experiment temperature is maintained by a constant-temperature convection oven. Fluid pressures at the inlet and the outlet are controlled by back-pressure regulators. In- and outlow fluid pressures were measured by KELLER® pressure sensors and the pressure readings were protocolled by a custom-made Labview® routine. Fluid samples can be retrieved for chemical analyses episodically from the outlfow line without disrupting the conditions within the reaction cell. This procedure allows the experimentalis to identify the point in time when condition within the cell are at steady-state and compute saturation states with regards to different actual and hypothetical solid reaction products.

During the ongoing percolation experiment, the percolation cell is located inside the oven, and recharge- and discharge fluids are connected by PEEK capillary tubing (Fig. 1C). The pump system, recharge- and discharge fluid strorage, and the board with the flow line are installed on a mobile rack. To facilitate the transfer of the percolation experiment from the oven laboratory to the X-ray microtomography laboratory for scanning, the mobile rack is equipped with an uninterruptible power supply.

## 2.2 Sample materials

The following samples have been chosen for intitial test runs aimed at demonstrating the unbiased X-ray observations of temporal changes in rock samples mounted in shrink-sleeves within the percolation cell. To assess differing states of pore fillings (air vs. fluid), a quartz-dominated sandstone (locality: Gildehaus, Romberg quarry, Germany; formation: Valangin, Lower Cretaceous) has been partially saturated with water. To demonstrate the sensitivity of in situ cell X-ray observations with respect to subtle differences in mineralogy and delicate textures, we mounted and scanned a mineralogically diverse tertiary sandstone ("Idaho Gray sandstone", Idaho, USA; Idaho formation, Tertiary). This sandstone was previously characterized by Halisch and Kaufhold (2015, pers. comm.) as coarse grained sandstone with more than 75 % of the grains ranging between 0.35 and 1.1 mm in size. It consists mostly of quartz (~ 80 %) besides heavily weathered muscovite (~ 11





%) and K-feldspar (~4 %). For a grain fraction < 100 µm, the mineral proportions identified by Halisch and Kaufhold (2015, pers. comm.) in these investigations revealed less quartz, but higher proportions of feldspar (~ 14 %), muscovite (~ 6 %) and a high amount of heulandite (~ 27 %).

As a test bed for a 4D study, the conversion of gypsum single crystals (selenite, Morocco; Mineraliengrosshandel Hausen

GmbH, Telfs, Austria) to anhydrite has been investigated by concomitant µ-CT monitoring of two percolation experiments over the course of 77 and 35 days. The experimental conditions were 110 °C, 45 bars fluid pressure, and a flow rate of 0.05 ml/min of a partially gypsum-saturated fluid (1.5 g/l dissolved gypsum; saturation state is 2 g/L).

**2.3 X-ray microtomography surveillance**

The X-ray microtomography scans aimed at assessing the applicability of the percolation cell were performed using a CT-

ALPHA system (ProCon, Germany) at the Department of Geosciences, University of Bremen, Germany. The sandstone micro-cores mounted in shrink-sleeves inside the percolation cell (Fig. 1) were scanned with a beam energy of 100 kV, an energy flux of 300 µA, and using an aluminium filter in 360°-rotation scans conducted with a step size of 0.225°. The gypsum (selenite) crystals were mounted in a simialr fashion and scanned with a beam energy of 130 kV, an energy flux of 350 µA, and a copper filter, using a 360° rotation with a step size of 0.3°. Reconstruction of the spatial information on the

linear attenuation coefficient in the samples was done with the Fraunhofer software VOLEX, using a GPU-hosted modified Feldkamp algorithm based on filtered backprojection (Feldkamp et al., 1984). Filtering of the raw data, volume reconstruction, segmentation, and rendering were done using Avizo 9.0.1 (FEI).

**3 Results**

Our results demonstrate that the placement of the micro-cores in the PEEK flow-through cell setup, mounted in shrink-

sleeves, does not bias the results of X-ray investigation and allows the concomitant surveillance of hydrothermal mineral-fluid interactions by repeated µ-CT scans without dismantling the sample. This assessment is based on data collected for a set of selected samples presented in following sections, which demonstrate the power of the developed exterimental setup for facilitating investigations of states and fabrics of water-rock systems undergoing active reaction.

**3.1 Detection of different states of pore filling in the mounted experiment**

The distinction between air-filled and fluid-filled pores of the partially water-saturated quartz-rich Gildehaus sandstone (Fig. 3) is straightforward. Air-filled pores (black in Fig 3) can easily be distinguished from fluid-filled pores (dark gray) in the unfiltered image (Fig. 3A, voxel size is 8.95 µm). Filtering of the data set (with non-local means 2D, Avizo; see Fig. 3B) allows definite segmentation of the different pore fillings and the rock material by means of gray level thresholding.





### 3.2 "Region-of-interest" tomography of parts of a mounted sample which exceeds the detector field of view

The reconstructed images of Idaho Gray sandstone sample (Ø 19 mm), scanned inside the assembled cell, surrounded by the PEEK cell material and a shrink-sleeve, clearly show the characteristic X-ray attenuation of the individual phases (Fig. 4). Aiming for an optimal resolution, the field of view for this scan covered approximately half the length of the sample and

roughly two thirds of the sample diameter. Delicate fabric elements, such as weathered muscovites and feldspars, can be recognized (voxel size is 6.22 μm) already in the unfiltered reconstructed images (Fig. 4A). Moreover, the recognition of structures, such as filigree zeolitic crystals (Fig. 4B and C, both filtered with 3D Sigma followed by 2D non-local means) overgrowing weathered muscovite demonstrates the suitability of the cell construction material for high resolution "region-of-interest tomography" (scan and reconstruction of "large" objects exceeding the detector field of view). In this regard, the

highest possible magnification will be limited by geometrical issues arising from the percolation cell dimensions and the size of the X-ray μ-CT scanner housing.

### 3.3 Concomitant X-ray surveillance of reaction progress without unmounting or dismantling the sample

To assess the experimental performance of the flow-through system in terms of the feasibility of repeated, concomitant scans of long-term percolation experiments, we monitored the gypsum / anhydrite transition as a case study of a 4D fabric

evolution. Two percolation experiments were performed (experiment FTGy-A for 77 days and experiment FTGy-B for 35 days) using artificially fractured selenite crystals that were subjected to a partially gypsum-saturated fluid (1.5 g/l), percolating at a flow rate of 0.05 ml/min at 110 °C and 45 bars (4.5 MPa) fluid pressure. To enhance the formation of flow paths within the selenite micro-core, pure $H_2O$ was initially used as recharging fluid in the experiment. Figure 4 shows exemplary μ-CT scans of experiment FTGy-A (voxel size is 23.18 μm) that document the reaction progress from (i) the

initially fractured micro-core (Fig. 5A), via (ii) the permeability-enhanced stadium after 7 days of percolation of pure $H_2O$ (Fig. 5B), to (iii) an almost final state with most of the gypsum starting material converted to anhydrite after further 70 days of percolation of the gypsum pre-saturated experimental fluid (Fig. 5C). 3D-analyses reveal that the conversion of gypsum to anhydrite (see illustration in Fig. 5D) was accompanied by an increase of porosity from 3.6 vol.% to 27.7 vol.% from the permeability-enhanced state (ii) (Fig. 5B) to the final state (iii) (Fig. 5C).

By concomitant μ-CT surveillance, a correlation between anhydrite growth and the formation and evolution of flow paths can be documented. In the course of the second percolation experiment FTGy-B (35 days, see Fig. A1, online supplementary material), a series of eight μ-CT scans were performed at several time steps (reaching from 0 to 35 days after the beginning of the experiment, as denoted in Fig. A1). The formation and broadening of a main cavity ("worm hole") is clearly visible from the set of successive volume reconstructions. The individual μ-CT slices for each time step reveal the continued growth

of anhydrite needles within the selenite. The nuclei of the newly formed anhydrite are located in apparent association with fractures in the selenite (see Fig. A1, online supplementary, stage after 10 days).





A comparison of fabric and mineralogy produced in both selenite percolation experiments provides additional information on the kinetic impact of different cooling rates on the mineralogy of the final run product. While experiment FTGy-A underwent fast cooling (< 5 min) from oven to room temperature before scanning, run FTGy-B was subjected to a slow cooling process (to room temperature within ca. 60 to 80 min) before each scan. The final fabric in both experiments features

densely intergrown anhydrite needles separated by fluid-filled intergranular space. The picture obtained after cooling dependends on the cooling rate: the newly formed anhydrite is apparently preserved after cooling in the fast-cooling run TGy-A. In contrast, the slow cooling of run FTGy-B resulted in an extensive secondary conversion of the newly formed anhydrite needles to gypsum.

## 4 Conclusion

The X-ray transparent reaction cell setup presented in this work holds great potential for fostering the knowledge about the interdependencies between changes in porosity distribution, permeability and their feedback relationship in the course of dissolution and precipitation reactions. μ-CT-based surveillance of percolation experiments can contribute in multiple ways to achieving a more sophisticated understanding of fluid-rock interactions:

(1) the consideration of true 3D spatially resolved μ-CT data of real pore space geometries will facilitate the formulation of relevant pore space models for transport reactions,
(2) the identification of preferred growth- or dissolution sites can be correlated to preferential fluid pathways,
(3) analysis of changing fluid compositions, changing mineral chemistry and volume proportions, can be used for the parametrization of coupled reaction-transport models,

(4) the 4D evolution of rock fabric and mineralogy, in particular the dissolution or precipitation rate spectra derived from the experiment by μ-CT-based monitoring, can be treated as benchmarks for computer models and can be used for parametrizing such coupled models with increased predictive power.

The results presented in this communication suggest that PEEK as the cell construction material is easily processable,
mechanically stable, and generally well-suited for X-ray imaging purposes. The straightforward cell design presented here is capable of manifold adaptions to specific experimental requirements in order to mimic a plethora of environmental and geotechnical conditions.

*Acknowledgements*. We are grateful to the people that have contributed to this development: first of all to Georg Nover for his qualified and cordial discussions concerning cell construction details. We thank Norbert Schleifer & Volker Fendrich and Robert Hinkes & Volker Feeser for helpful insights into their laboratory materials and methods. We are greatly indebted to





Elke Sorgenicht, forewoman of the mechanical workshop, and her staff for their brilliant technical realization. We thank
Matthias Lange for his help with flow path issues and Labview programming. Thanks also go to Martin Kölling and Katja
Beier for discussions in the early phase of this project. Matthias Halisch and Stephan Kaufhold are acknowledged for
providing both sample and characterization of Idaho sandstone, Norbert Schleifer is thanked for providing an aliquot of
Gildehaus sandstone.

This study was funded through a grant of the DFG to WB within a Reinhart-Koselleck Project BA 1605/10-1.

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




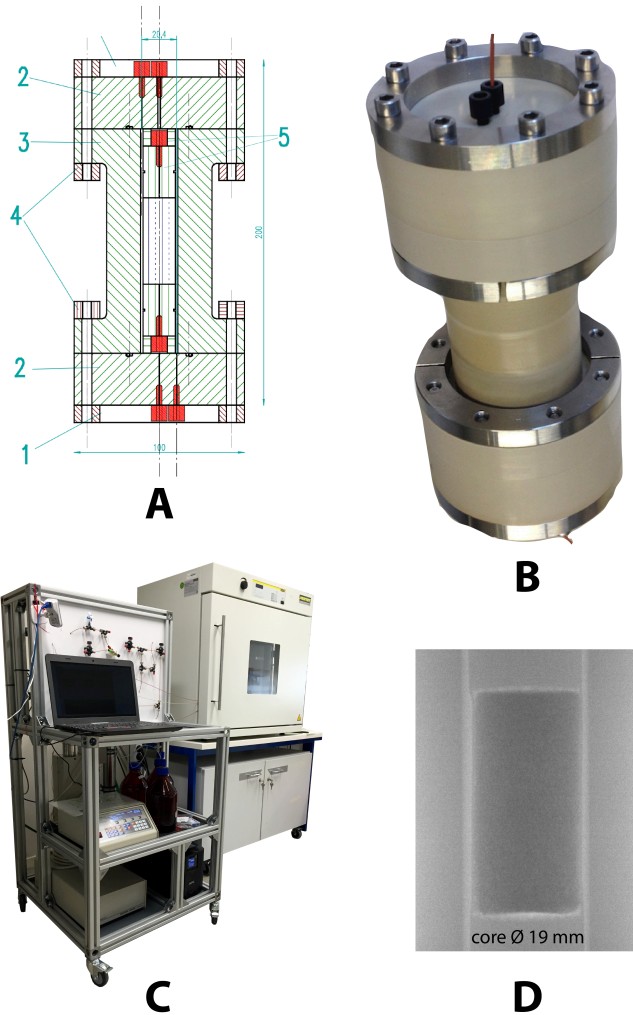

**Figure 1. X-ray transparent PEEK flow-through reaction cell. (A) Schematic illustration of the percolation cell, the location of the core is indicated. Note: the different parts of the assembled cell are labelled as follows: 1–steel rings, 2–end caps, 3–central part, 4–steel half-rings, 5–outlet plug of the core holder and 10–32 fingertight fitting. (B) Photograph of the assembled cell, with PEEK tubing 1/16" (1.60 mm) OD attached. (C) Photograph of an ongoing percolation experiment: the percolation cell is located inside the oven, recharge- and discharge fluids are connected by PEEK capillary tubing. The pump system, recharge- and discharge fluid strorage, and the board with the flow line are installed on a mobile rack. To facilitate the transfer of the percolation experiment from the oven laboratory to the X-ray microtomography laboratory for scanning, the mobile rack is equipped with an uninterruptible power supply. (D) μ-CT transmission image of a gypsum core mounted inside the percolation cell.**





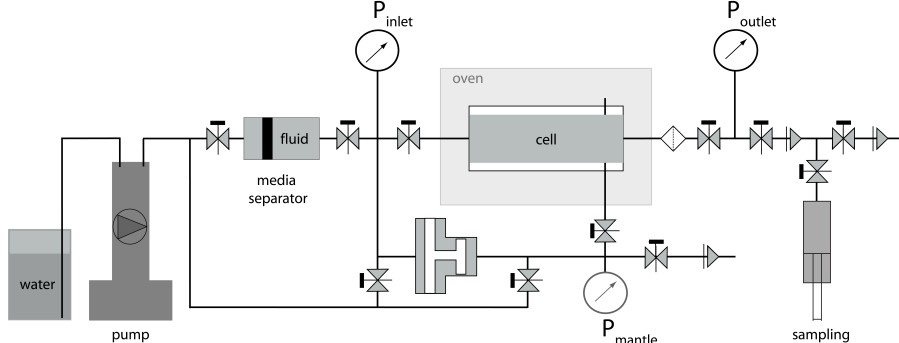

**Figure 2. Scheme of the flow line. All tubings, valves etc. are made of PEEK, the pins of the needle valves are made of titanium.**





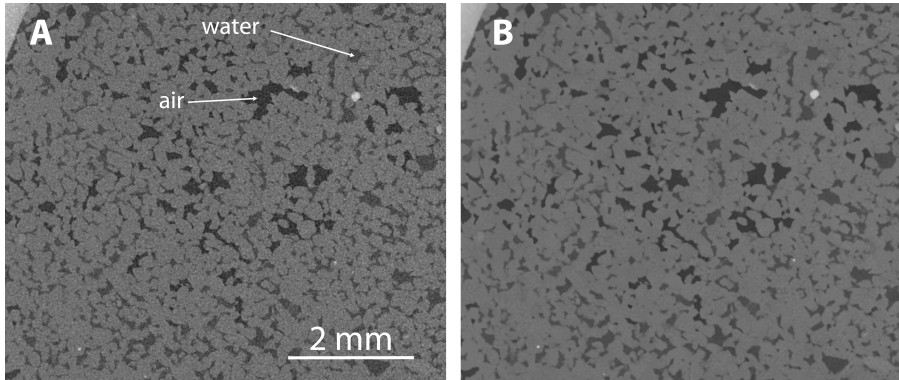

**Figure 3. Reconstructed images of a partially saturated, quartz-rich sandstone sample (Gildehaus, Romberg quarry; Ø 19 mm) demonstrating the virtually non-existent interference of the PEEK cell setup with distinctive X-ray observations. The samples are**
5 **scanned inside the assembled cell, surrounded by the PEEK cell material and a shrink-sleeve. (A) Unfiltered reconstruction. The distinction between air-filled (black) and fluid-filled (dark grey) pores is already straightforward (voxel size is 8.95 µm). (B) The reconstructed image, treated with 2D non-local means filter, allows definite segmentation of the pore fillings. Note the shrink-sleeve in the upper left of each image.**



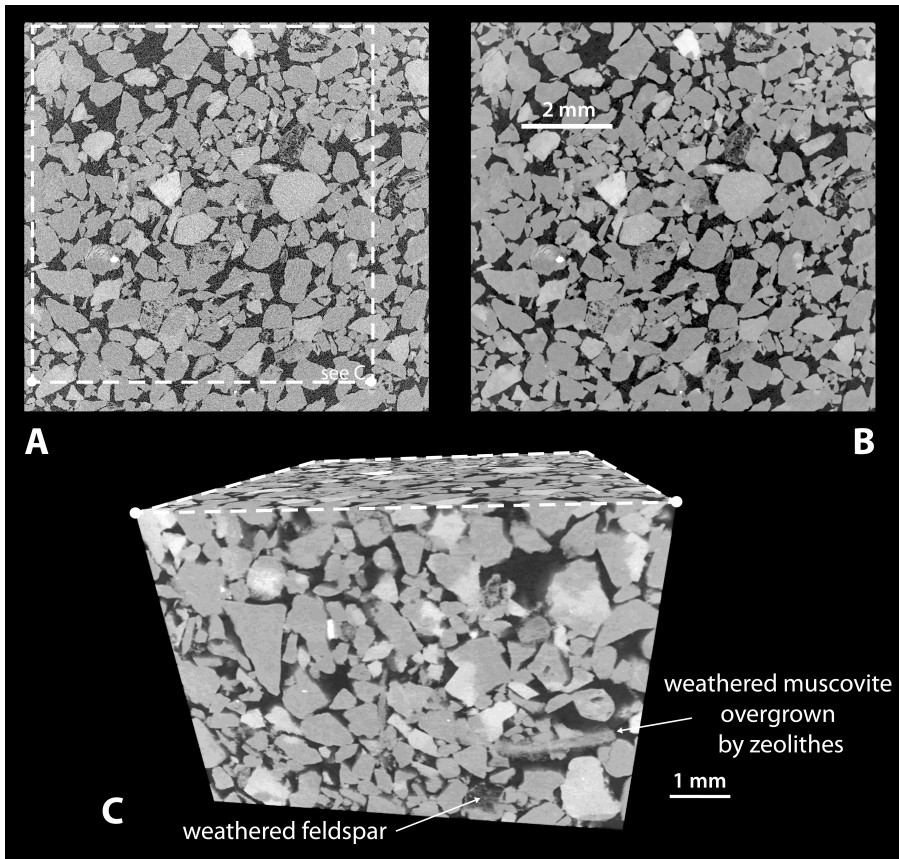

**Figure 4. Reconstructed images of a micro-core of Idaho Gray sandstone (Ø 19 mm), scanned inside the assembled cell, surrounded by the PEEK cell material and a shrink-sleeve. (A) Unfiltered reconstruction. Delicate fabric elements such as weathered muscovites and feldspars can be recognized (voxel size is 6.22 μm). (B) The filtered image (3D Sigma followed by 2D non-local means, both Avizo 9.0.1) of the same slice. (C) 3D volume reconstruction, with the area defined in (A) being the top layer. The recognition of structures such as filigree zeolitic crystals overgrowing weathered muscovite impressively demonstrates the suitability of the cell construction material for "region-of-interest tomography" (scan and reconstruction of "large" objects exceeding the detector field of view).**



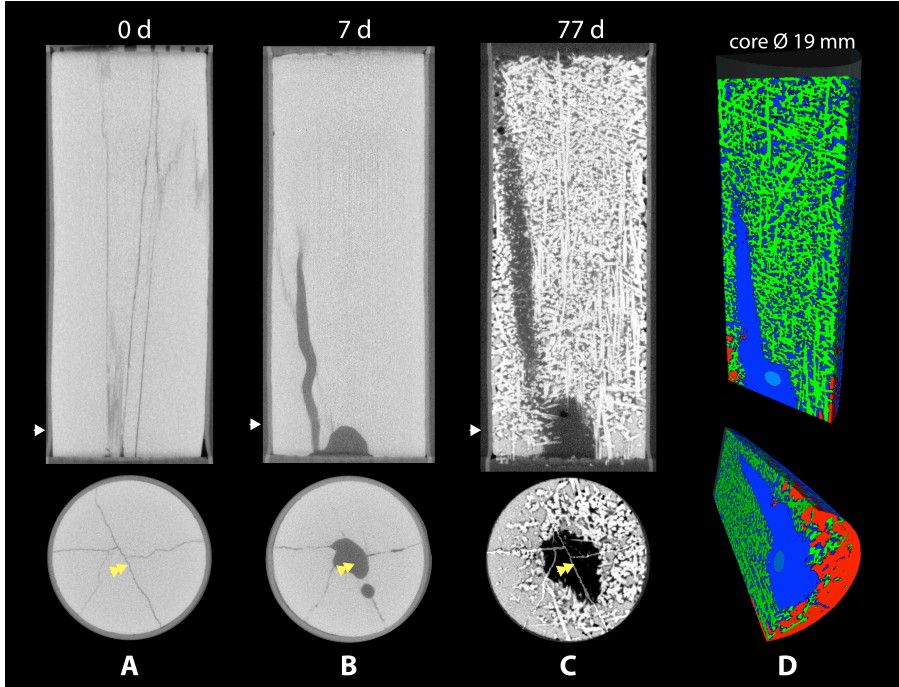

**Figure 5. 4D series of the conversion from gypsum to anhydrite (110 °C, fluid pressure 45 bars (4.5 MPa), direction of flow: upward). In figures A-C the cross sections are cut at slightly different depths (voxel size is 23.18 μm). (A) Gypsum single crystal (selenite, Morocco), initial state, artificially fractured. (B) Intermediate state, after 7 days percolation of gypsum-undersaturated water (pure H2O) to increase permeability. (C) Late stage of experiment after further 70 days of percolation of a partially gypsum-saturated fluid (1.5 g/l). Gypsum has almost entirely been replaced through growth of secondary anhydrite needles. (D) 3D model of segmented pore space, anhydrite and gypsum. Note: (i) white arrowheads in A-C, annotated to the core cross section, mark the position of the single slice shown below; (ii) yellow double arrowheads in A-C mark the position of the initial artificial fracture within the slice.**



**Figure A1. Documentation of formation and evolution of flow paths in the course of gypsum dissolution and anhydrite growth by concomitant μ-CT surveillance. A series of eight μ-CT scans were performed at several time steps (reaching from 0 to 35 days after the beginning of the experiment, as denoted in the figure). From the volume reconstructions the formation and broadening of a main cavity ("worm hole") is clearly visible (voxel size is 23.18 μm). The individual slices disclose the continued growth of anhydrite needles within the selenite. The nuclei of the newly formed anhydrite are located in association with fractures in the selenite (see state after 10 days). Note: the orange plane within the volume reconstruction marks the level of the single slice shown below.**