# Peer review of "A new X-ray transparent flow-through reaction cell for a $\mu$ -CTbased concomitant surveillance of the reaction progress of hydrothermal mineral-fluid interactions"

_Solid Earth, 2016_

## Referee Comment (RC1) · G. Nover (Referee) · 29 Feb 2016

Technical corrections: Mistype on page 5 line 13 'similar'

Comments:

It was a pleasure for me to read this paper.

The paper is well organized, clearly written and follows new aspects in understanding rock-fluid interactions in more detail. The technical section clearly outlines the idea behind and describes its realization in a very sophisticated experimental setup. Relevant

publications are considered and critically discussed, highlighting the unique concept of the authors. They follow new ideas in experimental laboratory work and open a new window for a more sophisticated understanding of fluid-rock interactions by means of 4D in-situ measurements.

The paper should be accepted as it is.

―――――――――――――――――

---

## Referee Comment (RC2) · HS Steeb (Referee) · 7 Mar 2016

**A new X-ray transparent flow-through reaction cell for a $\mu$ -CT- based concomitant surveillance of the reaction progress of hydrothermal mineral-fluid interactions**

Wolf-Achim Kahl, Christian Hansen, Wolfgang Bach

The manuscript presents a new X-ray transparent cell concept for reactive transport processes of reservoir rocks. The topic of the paper is very interesting as XRCT (X-Ray Computed microTomography) is well available in our days and advanced imaging techniques give a great inside into morphology evolution for complex processes like the one mentioned here. The content of the manuscript fits therefore very well in the Special Issue of SE.

Nevertheless, my impression is that various important aspects keep unclear in the manuscript. Besides various missing technical details, the reviewer mainly misses detailed images and/or technical drawing explaining the cell concept and the underlying "imaging" procedure. Nevertheless, this could be easily improved in a revised version. The reviewer is looking forward to that version!

Recommendation: Major Revision

Remarks in detail:

- To discuss differences of your cell concept compared to others available in the literature for similar applications, it would be very important to cite additional papers, where X-ray transparent cells and/or feasibility studies have been discussed extensively, cf. [1,2] and others. Further, a critical discussion about advantages/disadvantages of your concept would be very much appreciated!
- Page -2-, Line 7:

In your mentioned workflow, all bench-top and synchrotron microtomography facilities are meant to be X-ray-based. This should be clarified. You could write e.g.: bench-top and synchrotron X-ray computed microtomography: micro-CT.

- Please check carefully the reference list (typos), i.e. replace Kesten with Kersten
- Page -2-, Line 7:

Why do you call mercury porosimetry a pure statistical consideration which is inferior to micro-CT? Mercury porosimetry is based on pore-scale physics (Washburn-Equation). On the one hand, it is true, that only an averaged pore size could be determined but on the the other hand, the spatial resolution of mercury porosimetry (<< 1 micrometer) is (much) higher then the resolution of various imaging techniques, like e.g. bench-top microtomography. Please clarify!

- Please provide materials properties of PEEK; this could be helpful for the reader. Further, I
  would suggest to discuss/compare X-ray transparency of PEEK especially w.r.t. alternative
  materials used for X-ray transparent cells (quartz glass, pure aluminium, carbon,...)
- It would be interesting for the reader if you share more advanced drawings of your cell and of the underlying concept. Perhaps you could provide CAD drawing (supplementary data or explosion sketch for mounting purposes) or detailed drawing of the sealing part? Further, a detailed list of parts (fittings, o-rings, etc) would also be of great interest. In detail: I miss especially a detailed explanation of the (upper) sealing part of the cell. I understand that the dumbbell part is mounted via o-rings to the upper part of the cell. How are the end-caps connected to the fluid lines? I see two 10-32 fingertight fittings, which I could not understand form the drawing. Do you use a dynamometric key to fix the upper part of the cell to the dumbbell part?

**se-2016-25**

- Page -3-, Line 28: You mean perhaps: 210 mm in height and 100 mm in diameter?
- Please provide more information about the used back pressure regulator (type, manufacturer, etc)
- How do you measure temperature when the sample is removed from the oven for imaging purposes? You do not have any thermocouple inside the cell?
- Please provide more details about the pressure sensors (type, pressure range, active/passive sensor). How large is the dead volume of the sensors? Have you thought about novel in-line pressure sensors used for instance in micro-fluidic devices?
- Could you provide more information about the dead volume of the total system (incl. dead volume of pressure sensor, valves, end-caps, etc) especially compared to pore volume? This could be of great importance when you change the fluid...
- Page -4-, Line 16:

It is very interesting and also quite important, that fluid could be extracted from the sample during the experiment. How exactly could that be done in your case? Could you guarantee (during the fluid sampling procedure) that the pore pressure state is kept constant? You are measuring pore pressure, please try to provide information.

- I understand, that the flow-through experiment is in the oven and only removed for imaging purposes. If this is done various times, how do you solve the registration problem of the cell? You are working with a voxel size of 8.95 micrometer, cf. Fig. 3). Thus, you need a higher precision for registration. Does that causes problems?
- Page -4-, Line 29:

Please describe in detail what you mean with partially saturated with water. How do you inject water in order to guarantee a partial saturation condition? Before injecting water, is the sample saturated with air? Do you flush the sample with CO2? Do you apply a vacuum? I think, even in a feasibility study this workflow information is important.

• Page -6-

I understand, that you would like to present a feasibility study discussing especially the cell and the flow-through concept for imaging. But even with this goal, I have a further point which could be included in the manuscript without much effort: During the flow-through experiment you measure the pressure gradient (please provide information...) according to Fig. 2. In addition, the flow-through experiment is fluid-flux-controlled. Thus, you could easily calculate permeability and even more, you know temporal permeability evolution through the time period of the experiment. Wouldn't it be interesting, especially in a feasibility study, to compare the exp. obtained permeability evolution with calculated permeability evolution from porosity evolution (i.e. Kozeny-Carman - which you got through the micro-CT snapshots)?

**References:**

- Fusseis, F., Steeb, H., Xiao, X., Zhu, W.-L., Butler, I. B., Elphick, S., & Mäder, U. (2014). A lowcost X-ray-transparent experimental cell for synchrotron-based X-ray microtomography studies under geological reservoir conditions. J Synchrotron Rad, 21, 251–253. http://doi.org/10.1107/S1600577513026969/pp5039sup2.zip
- Ott, H., de Kloe, K., van Bakel, M., Vos, F., van Pelt, A., Legerstee, P., et al. (2012). Core-flood experiment for transport of reactive fluids in rocks. Review of Scientific Instruments, 83(8), 084501. http://doi.org/10.1063/1.4746997

---

## Author Comment (AC1) · 21 Mar 2016

Deer Reviewers Georg Nover and Holger Steeb,

we received your reviews and want to thank you for your comments, which helped us greatly to improve our manuscript.

Introductorily, we want to state the following:
*As we had in mind when this publication was set up as a method note, we plan to share the information of the cell construction as more detailed photograph (in the form of an exploded view) and a table with a parts list.*

*However, we hope that the reviewers understand that we do not intend to present these details in a manuscript that is under review in an open discussion format, not yet accepted for publication.*

In the following, we respond to the comments of Holger Steeb:

• *To discuss differences of your cell concept compared to others available in the literature for similar applications, it would be very important to cite additional papers, where X-ray transparent cells and/or feasibility studies have been discussed extensively, cf. [1,2] and others. Further, a critical discussion about advantages/disadvantages of your concept would be very much appreciated!*
**We thank Holger Steeb for his hint to additional papers that feature X-ray transparent cells. We included those and another reference in our introductory section and finetuned the delineation of the particular demands of the rock-water systems we are aiming at with our cell setup.**

• *Page -2-, Line 7:*
*In your mentioned workflow, all bench-top and synchrotron microtomography facilities are meant to be X-ray-based. This should be clarified. You could write e.g.: bench-top and synchrotron X-ray computed microtomography: micro-CT.*
**To avoid ambiguity, we adopted the expression of Cnudde & Boon (2013): laboratory-based and synchrotron-based μ-CT.**

• *Please check carefully the reference list (typos), i.e. replace Kesten with Kersten*
**Thanks. One should never trust references in published papers and copy/paste them carelessly...**

• *Why do you call mercury porosimetry a pure statistical consideration which is inferior to micro-CT? Mercury porosimetry is based on pore-scale physics (Washburn-Equation). On the one hand, it is true, that only an averaged pore size could be determined but on the the other hand, the spatial resolution of mercury porosimetry (<< 1 micrometer) is (much) higher then the resolution of various imaging techniques, like e.g. bench-top microtomography. Please clarify!*
**Mercury porosimetry is an extremely useful (albeit contaminant) technique, which provides important information about the porosity of samples, as it covers pore sizes over a range of 5 orders of magnitude from 0.4 mm to less than 4 nm. However, the results are also limited in several ways (for a detailed discussion see e. g. Giesche, 2006). The modified Young-Laplace equation, which commonly referred to as the Washburn equation, assumes a cylindrical pore geometry. The real pore shape is however quite different and the cylinder pore assumption can lead to major differences between the analysis and reality. We mean "statistical" in the sense that mercury porosimetry determines the largest entrance to a pore, but not the actual pore size and shape.**

*• Please provide materials properties of PEEK; this could be helpful for the reader. Further, I would suggest to discuss/compare X-ray transparency of PEEK especially w.r.t. alternative materials used for X-ray transparent cells (quartz glass, pure aluminium, carbon,...)*

**Good remark, we will report the following selected material properties of PEEK: density, yield stress/tensile strength, and heat deflection temperature).**
**We chose PEEK as cell material for two reasons: (1) because of the good experience colleagues made when they integrated caps or other small pieces made of PEEK into their usual metal-based flow cell setups; (2) We were delighted to find that our workshop has experience in handling PEEK as construction material. At this point, we do not see the necessity to start a more theoretical search for other materials than PEEK which is what we favored, which imaging properties are shown to be well-suited for µ-CT surveillance, as shown by our manuscript.**

*• It would be interesting for the reader if you share more advanced drawings of your cell and of the underlying concept. Perhaps you could provide CAD drawing (supplementary data or explosion sketch for mounting purposes) or detailed drawing of the sealing part? Further, a detailed list of parts (fittings, o-rings, etc) would also be of great interest. In detail: I miss especially a detailed explanation of the (upper) sealing part of the cell. I understand that the dumbbell part is mounted via o-rings to the upper part of the cell. How are the end-caps connected to the fluid lines?*

**As we had in mind when this publication was set up as a method note, we plan to share the information of the cell construction as more detailed photograph (in the form of an exploded view) and a table with a parts list.**
**However, we hope that the reviewers understand that we do not intend to present these details in a manuscript that is under review in an open discussion format.**

*I see two 10-32 fingertight fittings, which I could not understand form the drawing. Do you use a dynamometric key to fix the upper part of the cell to the dumbbell part?*

**All fittings are fixed finger-tight, the metal rings can be fixed hand-tight without a dynamometric key.**

*• Page -3-, Line 28:*
*You mean perhaps: 210 mm in height and 100 mm in diameter?*

**true**

*• Please provide more information about the used back pressure regulator (type, manufacturer, etc)*

**We use VICI backpressure regulators (JR-BPR2) which is intended for the range of 20-103 bar**

*• How do you measure temperature when the sample is removed from the oven for imaging purposes? You do not have any thermocouple inside the cell?*

**In consideration of the duration of the scan we take the temperature reading from the inside of the scanner's cabinet.**

*• Please provide more details about the pressure sensors (type, pressure range, active/passive sensor). How large is the dead volume of the sensors? Have you thought about novel in-line pressure sensors used for instance in micro-fluidic devices?*

**Currently we use KELLER series 33X pressure sensors (floating piezoresistive transducer; range 0-100 bar; material in contact with media: Hastelloy, Viton) with a dead volume change reported to be < 0.1 mm³. The design of both cell and flow line will be continiuously developed (e. g., it will be augmented by differential pressure transmitters with suitable pressure ranges which will emerge from the first experiments). However, all modifications have to be capable**

to withstand the extreme conditions in terms of pH we expect from the chemical systems we will investigate

*• Could you provide more information about the dead volume of the total system (incl. dead volume of pressure sensor, valves, end-caps, etc) especially compared to pore volume? This could be of great importance when you change the fluid...*

We are not sure if we understand the term "dead volume" in respect to fluid change. If "dead volume" refers to extra-column volumes (not properly installed fittings etc.), a system will be particularly susceptible when using narrow internal diameter columns, as extra-column volumes become a larger percentage of the total volume. However, the tubing with 500 μm diameter we use does not fall in this range. Does the term 'dead-volume' refer to volumes within the flow line which are not swept by the mobile phase? In any case, we have to ask: when is dead volume a real concern?

The duration of our experiments measures in weeks or months. In our experiments we are interested in the chemical changes concerning mineralogy and fluid chemistry. Fluids are sampled daily to weekly between the scans. Mass balance calculations will rely on the volumes of pores and rock (as determined by μ-CT), and on the amount of fluid that passed the cell. All fluids will pass the cell, and the amount is logged by the readings of the pump system. If we change fluids during an experiment, the most interesting "dead volumes" will be within the sample in terms of dead-end pores or zones of reduced flow.

*• Page -4-, Line 16:*
*It is very interesting and also quite important, that fluid could be extracted from the sample during the experiment. How exactly could that be done in your case? Could you guarantee (during the fluid sampling procedure) that the pore pressure state is kept constant? You are measuring pore pressure, please try to provide information.*

The extraction of fluids is shown in Figure 2: we use a syringe to recover fluid that has left the cell. The fluid sampling device is positioned after a backpressure valve. If gas content / solubility is an issue, the fluid extraction unit will be located between two backpressure valves to establish both separation from the cell and maintainance of pressurization during fluid extraction.

*• I understand, that the flow-through experiment is in the oven and only removed for imaging purposes. If this is done various times, how do you solve the registration problem of the cell? You are working with a voxel size of 8.95 micrometer, cf. Fig. 3). Thus, you need a higher precision for registration. Does that causes problems?*

For registration we use the Avizo 9.0.1 software. Initially we perform a manual pre-alignment within the functionality of the transform editor, using the spiderweb groove pattern in the fluid inlet of the core holder end-cap as landmarks. In the subsequent workflow within Avizo it is possible to resample the dataset. Depending on the actual differences of the 4D states (e. g., a simple crack vs. complex, inhomogeneous dissolution patterns), it may not always the best choice to proceed from point-based to intensity-based registration techniques to attain a satisfactory result.

*• Page -4-, Line 29: Please describe in detail what you mean with partially saturated with water. How do you inject water in order to guarantee a partial saturation condition? Before injecting water, is the sample saturated with air? Do you flush the sample with CO2? Do you apply a vacuum? I think, even in a feasibility study this workflow information is important.*

To create a sandstone sample that features both water- and air-filled pore space, we first flooded the sample completely with water at 50 bar fluid pressure, and then depressurized and disconnected the tubing to permit evaporation towards the state we eventually scanned.

*• Page -6-I understand, that you would like to present a feasibility study discussing especially the cell and the flow-through concept for imaging. But even with this goal, I have a further point which could be included in the manuscript without much effort: During the flow-through experiment you measure the pressure gradient (please provide information...) according to Fig. 2. In addition, the flow-through experiment is fluid-flux-controlled. Thus, you could easily calculate permeability and even more, you know temporal permeability evolution through the time period of the experiment. Wouldn't it be interesting, especially in a feasibility study, to compare the exp. obtained permeability evolution with calculated permeability evolution from porosity evolution (i.e. Kozeny-Carman - which you got through the micro-CT snapshots)?*

**It will definitely be interesting to compare an experimentally obtained permeability evolution and a calculated permeability of a segmented 3D pore space model. We are looking forward to perform and present experiments that will investigate this correlation in the course of the future work.**

**References:**

Giesche, H. (2006) Mercury Porosimetry: a General (Practical) Overview. *Part. Part. Syst. Charact.* **23**, 1-11.